# Cardiovascular Disorder after Cardiotoxic Non-Hodking’s Lymphoma Treatment: A Case Report

**DOI:** 10.3390/medicina58040489

**Published:** 2022-03-29

**Authors:** Diana Žaliaduonytė, Rita Kleinauskienė, Gintarė Muckienė, Vytautas Zabiela

**Affiliations:** 1Department of Cardiology, Medical Academy, Hospital of Lithuanian University of Health Sciences, Eivenių St. 2, 50161 Kaunas, Lithuania; gintare.muckiene@gmail.com (G.M.); zabielav@gmail.com (V.Z.); 2Medical Academy, Lithuanian University of Health Sciences, Mickeviciaus St. 9, 44307 Kaunas, Lithuania; 3Kaunas Region Society of Cardiology, 50009 Kaunas, Lithuania

**Keywords:** aortic valve stenosis, coronary artery disease, heart failure, cardiotoxic treatment

## Abstract

The non-Hodgkin’s lymphomas are a diverse group of lymphoid neoplasms that collectively rank fifth in cancer incidence and mortality. Patients treated with mediastinal radiotherapy and/or anthracycline-containing chemotherapy are known to have increased risks of coronary heart disease, valvular heart disease, and heart failure. This may be the result of cancer treatment cardiotoxicity or may be due to accelerated development of cardiovascular disease. We presented 41-year-old male who was admitted to the hospital because of congestive heart failure. He has a medical history of non-Hodgkin’s lymphoma treated with anthracycline-based chemotherapy and mediastinal radiotherapy almost 20 years ago. Echocardiography showed significant aortic valve stenosis, thickened and fibrotic pericardium. Coronary angiography showed diffuse three-vessel coronary artery disease. The patient was referred for surgical treatment. Aortic valve replacement, coronary artery bypass grafting and pericardiectomy were successfully performed, symptoms of heart failure reduced.

## 1. Introduction

Cardiovascular diseases (CVD) are a group of disorders of the heart and blood vessels. Dyslipidemia, diabetes, hypertension, smoking, family history of heart and vascular diseases—well known cardiovascular risk factors. Patients with cancer usually receive combined treatment—cancer drugs and sometimes radiation, with the potential for cardiotoxic effects [1]. CVD after cancer treatment may be the result of direct cardiovascular lesion caused by the treatment itself or of expedited atherosclerosis, especially in the presence of traditional cardiovascular risk factors [2]. We present a case report of young man who suffered of advanced cardiovascular disease. He was treated for non-Hodgkin lymphoma (NHL) 20 years ago. After the treatment, the patient’s condition improved and symptoms of heart failure reduced.

## 2. Case Report

A 41-year-old male represented to the Emergency Department complaining of dyspnea, fatigue, orthopnea and growing weight. These symptoms had been worsening over the past 3 months. The patient was treated of NHL with 8 courses of anthracycline-based chemotherapy (CHT) (Doxorubicin 640 mg/m^2^, Cyclophophamide 640 mg, Vincristine 16 mg, Prednizolone 320 mg) and mediastinal radiation therapy (RT) (30 Gy) in 1999. After appropriate treatment remission of NHL has been achieved. Cardiovascular risk factors were obesity (body mass index 33 kg/m^2^), dyslipidemia, smoking (more than 20 cigarettes a day, more than 25 year), family history of CVD was unremarkable.

The patient‘s blood pressure was 110/60 mmHg, pulse rate—90 beat per minute, a respiratory rate—20 breaths per minute. During physical examination of the patient we found ascites, peripheral cyanosis, extremities’ oedema, obesity (body mass index 33 kg/m^2^). During cardiac auscultation established grade II systolic murmur at the heart apex and grade IV systolic murmur at aortic area. Lung auscultation revealed decreased vesicular breathing bilateral, especially on the right side.

NT-proBNP was more than 1500 ng/L, other blood tests were normal. Chest X-ray showed right side hydrothorax up to the sixth rib (Figure 1).

Transthoracic echocardiography (TTE) demonstrated fibrocalcinosis of aortic root, aortic valve annulus, aortic valve cusps (Figure 2a) and moderate to severe aortic stenosis (Vmax 3.36 m/s, Gmean27 mmHg) (Figure 2b), decreased left ventricular (LV) systolic function (ejection fraction 35%), thickened, fibrotic pericardium. In the presence of right ventricular (RV) pressure overload, the interventricular septum shifted towards the LV.

These findings were confirmed by transesophageal echocardiography (TOE), aortic valve is tricuspid with severe calcinosis, aortic valve area (AVA) was 1.1 cm^2^ (Figure 3). 

Coronary angiography showed extremely diffused three-vessel coronary artery calcification (Figure 4).

In a multidisciplinary heart team was decided to perform the aortic valve replacement (AVR) and coronary artery bypass grafting (CABG) surgery. The operation was complicated by bleeding, multiple mediastinal adhesions, the whole heart was fixed by a calcified pericardium, which led moderate heart constriction. Aortic valve was tricuspid with severe calcification of cusps with spreading of calcification to aortic annulus and aortic wall. The operation lasted 7 h where AVR with mechanical aortic valve prosthesis, CABG using four coronary artery junctions and pericardiectomy were performed. Acute postoperative period was complicated by renal failure, pneumonia, right side hydrothorax. In 41 days, the patient’s health improved, HF symptoms, renal insufficiency revealed and the patient was prescribed for rehabilitation treatment.

## 3. Discussion 

The non-Hodgkin lymphoma (NHL) is malignant neoplasms group originating from the lymphoid tissues. The NHL based treatment includes chemotherapy (CHT) and radiation therapy (RT) [1,2]. Anthracyclines efficacy in treating cancer is limited by a cumulative dose-dependent cardiotoxicity [3]. Survivors of aggressive NHL have a 17% incidence of clinical heart failure (HF) at 5 years [4]. At Doxorubicin cumulative dose of 400 mg/m^2^, there is a 5% risk of developing HF, which increases to 25–48% at 700 mg/m^2^ [5,6]. CHT cardiotoxicity can be two types. Type 1 is associated with cardiomyocyte death, necrosis or apoptosis, and as result is not reversible. Type 2 is based on dysfunction of cardiomyocyte and therefore may be reversible. Doxorubicin is assigned to the type 1 cardiotoxicity [5,6]. 

The cardiotoxicity can be acute, early and late. Acute toxicity develops in less than 1% of patients after first dose and is usually reversible [1,6]. Early cardiotoxic effect occurring within 1 year of exposure, while late cardiotoxic effect occurring 1 to 20 years after initial exposure [6], median of 7 years after treatment [1].

Irradiation-induced CVD occur in 10% to 30% of patients within 5–10 years following treatment [7]. Since CVD develops slowly, it is normally not seen before more than 15 years of follow-up [7,8]. In a study of 6039 NHL patients followed for a median of 9 years after RT, cardiovascular complications were seen in 11.6%, the most common being coronary heart disease (CHD) (19%), arrhythmia (16%), HF (12%), valvular heart disease (VHD) (11%), and pericardial disease (5%) [9]. When RT is combined with CHT, systolic dysfunction is generally observed [1,9]. There are established two mechanisms causing damage during RT: direct ionization of cell components and indirect radiolysis of intra-and extracellular water [7]. Mediastinal RT accelerates the atherosclerosis process, resulting early developing of CHD and VHD [7,8]. Pericardial involvement includes acute and chronic pericardial disease, pericardial effusion [8]. The patients treated with CHT and/or RT are at high risk of developing CVD and long-term clinical follow-up and testing for presence of HF, CHD or VHD, may be useful to identify patients with cardiac disease.

## 4. Conclusions

The patient was hospitalized for HF symptoms after previous cardiotoxic treatment. Unfortunately, during this period, the patient was not followed-up by cardiologist. Myocardial dysfunction and HF are the most concerning cardiovascular complications of cancer therapies and cause an increase in morbidity and mortality. An important method for preventing heart failure must be balanced between the anti-tumor efficacy of the treatment and its potential cardiotoxicity. Patient’s cardiovascular system and cardiotoxicity risk factors must be monitored before and after cancer treatment based on ECS guidelines of cardiotoxicity [1,10].

## Figures and Tables

**Figure 1 medicina-58-00489-f001:**
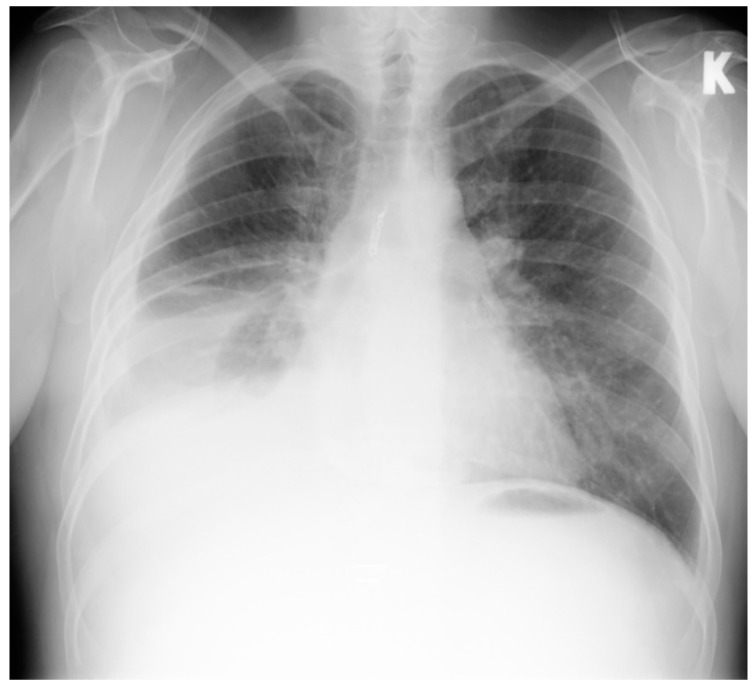
Chest X-ray. Right side hydrothorax. K-left side.

**Figure 2 medicina-58-00489-f002:**
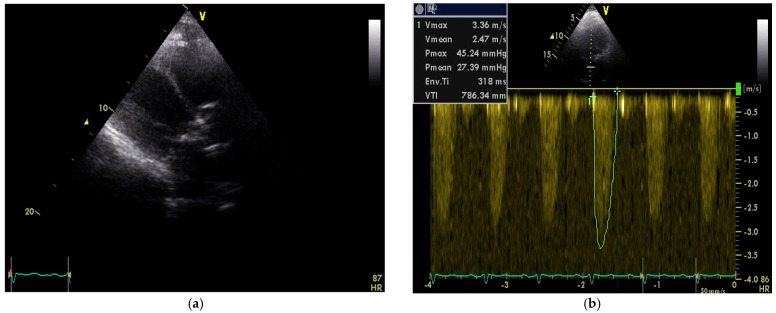
(**a**) Transthoracic echocardiography demonstrated fibrocalcinosis of aortic root, aortic valve annulus, aortic valve cusps and pericardium; (**b**) Peak velocity through aortic valve 3.36 m/s, mean gradient through aortic valve—27.39 mmHg.

**Figure 3 medicina-58-00489-f003:**
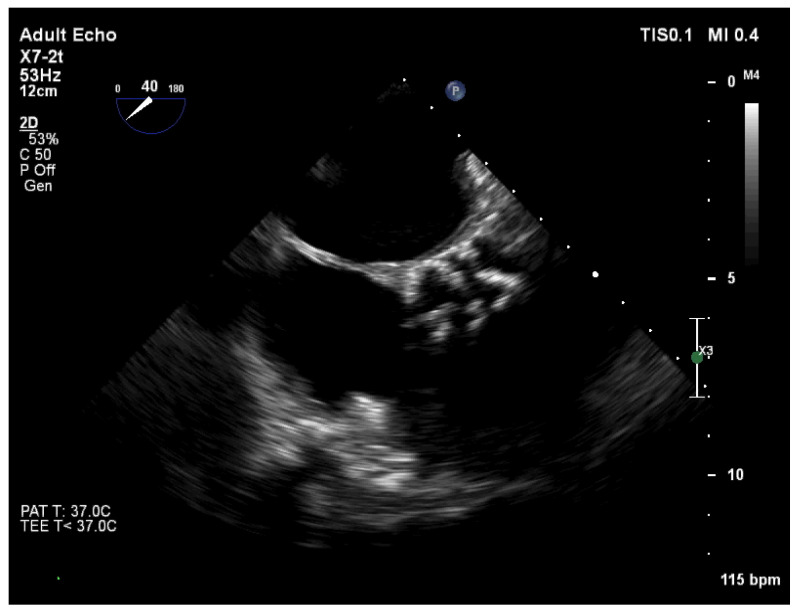
TOE: aortic valve is tricuspid with severe calcinosis. AVA was 1.1 cm^2^.

**Figure 4 medicina-58-00489-f004:**
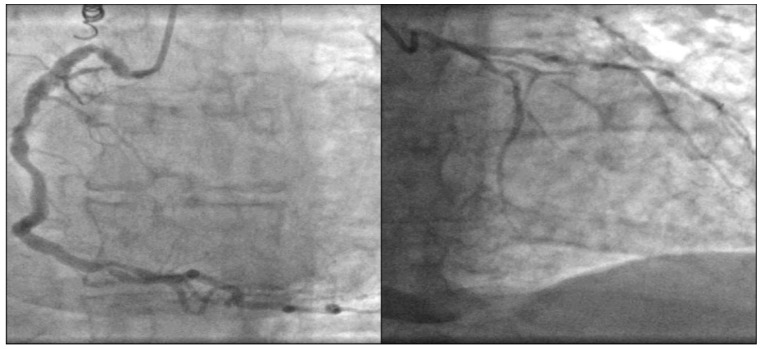
Coronary angiography. Diffused coronary artery calcification.

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
