# Peer review of "Cardiovascular Disorder after Cardiotoxic Non-Hodking’s Lymphoma Treatment: A Case Report"

_medicina, 2022, doi:10.3390/medicina58040489_

Round 1

Reviewer 1 Report

  1. The hypertension and diabetes should be added to the introduction of cardiovascular risk factors.
  2. As the patients with reduced ejection, the AS velocity, and gradient may be low,the effective aortic valve area or low dose dobutamine protocol should be combined with the evaluation of AS.
  3. TTE shows obvious calcification of the aortic valve, whether there is congenital aortic stenosis,such as a bicuspid aortic valveor a unicuspid aortic valve.Surgical records need to be presented.

Author Response

Response to Reviewer 1 Comments

Dear Reviewer,

Thank you for your comments and suggestions.

Point 1:  The hypertension and diabetes should be added to the introduction of cardiovascular risk factors.

Response 1:  We supplemented the introduction.

Point 2: As the patients with reduced ejection, the AS velocity, and gradient may be low,the effective aortic valve area or low dose dobutamine protocol should be combined with the evaluation of AS.

Response 2: We understand that left ventricular dysfunction can be associated with lower transaortic and LVOT velocities and therefore underestimation of gradient, so for this patient effective aortic valve area was measured by continuity equation principle. AVA was 1,1 cm2.

 We didn‘t done low dose dobutamine test, because patient had significant multi-vessel CAD requiring surgical revascularisation on the instant and together we had significantly calcificated tricuspid aortic valve.  In a multidisciplinary heart team was decided to perform coronary artery bypass grafting surgery and the aortic valve replacement.

Point 3: TTE shows obvious calcification of the aortic valve, whether there is congenital aortic stenosis, such as a bicuspid aortic valveor a unicuspid aortic valve.Surgical records need to be presented.

Response 3: We suplemmented surgical records part.  At this case were tricuspid aortic valve with marked calcification of aortic valve cusps, passing into the aortic valve anullus and aortic wall.

Reviewer 2 Report

Anthracycline based chemotherapy is the backbone of non Hodgkin's lymphoma treatment and has very good survival outcome so I do not anticipate that this could be potentially avoided as suggested/ recommended by the authors. Anthracycline and mediastinal radiation induced cardiovascular toxicities is  well established.  Routine baseline Echo and then serially every 3 months or earlier (if acute cardiorespiratory symptoms) in patient receiving anthracycline  based regimens  is standard of care in order to detect cardiotoxicities especially CHF early. If diagnosed appropriate dose modifications, temporary or permanent discontinuation can be done. Even  long term cardiovascular disease is well known. 

Also, the discussion is very brief and introduction is long. I would suggest that it should be other way round. the key message is not clear. I am not sure if the authors want to emphasize  lack of awareness to refer to cardiooncology by the oncologist. If yes, it should be supported with some data.  

Author Response

Dear Reviewer,

Thank you for your comments and suggestions.

Point1: Anthracycline based chemotherapy is the backbone of non Hodgkin's lymphoma treatment and has very good survival outcome so I do not anticipate that this could be potentially avoided as suggested/ recommended by the authors. Anthracycline and mediastinal radiation induced cardiovascular toxicities is  well established.  Routine baseline Echo and then serially every 3 months or earlier (if acute cardiorespiratory symptoms) in patient receiving anthracycline  based regimens  is standard of care in order to detect cardiotoxicities especially CHF early. If diagnosed appropriate dose modifications, temporary or permanent discontinuation can be done. Even  long term cardiovascular disease is well known. 

Also, the discussion is very brief and introduction is long. I would suggest that it should be other way round. the key message is not clear. I am not sure if the authors want to emphasize  lack of awareness to refer to cardiooncology by the oncologist. If yes, it should be supported with some data.  

Response 1:  We apologise that our key message was not so clear. We want to describe, that this patient was not controlled after oncologic treatment. We have guidelines but sometimes we have not well cooperative patient.

We changed introduction and discussions part based your suggestions. Also we correct conclusions part. We hope that now it will be more clearly.

Thank You for your revision.

Reviewer 3 Report

This is a case report on cardiovascular disorder after cardiotoxic non-Hodgkin’s lymphoma treatment. The authors reported that the patient recovered from the disorder after treatment and suggested the need for a balance between drugs that treat cancer and their potential side effects in causing other form of toxicity. I have minor concerns

Minor

  1. The introduction should end with a brief statement on the case study
  2. Lines 95&96 overlap with figure 3 legends

Author Response

Dear Reviewer,

Thank you for your comments and suggestions.

Point 1: The introduction should end with a brief statement on the case study

Response 1:  We  supplemented our introduction with brief statement on the case study.

Point 2: Lines 95&96 overlap with figure 3 legends.

Response 2:  We correct this overlap.

Round 2

Reviewer 2 Report

Thank you for revising the manuscript. It reads better with few minor grammatical errors.